# One Saddle Point and Two Types of Sensitivities within the Lorenz 1963 and 1969 Models

**Bo-Wen Shen** [1,*] , **Roger A. Pielke, Sr.** [2] **and Xubin Zeng** [3]

1   Department of Mathematics and Statistics, San Diego State University, San Diego, CA 92182, USA
2   Cooperative Institute for Research in Environmental Sciences, University of Colorado Boulder, Boulder, CO 80203, USA; pielkesr@colorado.edu
3   Department of Hydrology and Atmospheric Science, The University of Arizona, Tucson, AZ 85721, USA; xubin@arizona.edu
*   Correspondence: bshen@sdsu.edu

**Abstract:** The fact that both the Lorenz 1963 and 1969 models suggest finite predictability is well known. However, less well known is the fact that the mechanisms (i.e., sensitivities) within both models, which lead to finite predictability, are different. Additionally, the mathematical and physical relationship between these two models has not been fully documented. New analyses, along with a literature review, are performed here to provide insights regarding similarities and differences for these two models. The models represent different physical systems, one for convection and the other for barotropic vorticity. From the perspective of mathematical complexities, the Lorenz 1963 (L63) model is limited-scale and nonlinear; and the Lorenz 1969 (L69) model is closure-based, physically multiscale, mathematically linear, and numerically ill-conditioned. The former possesses a sensitive dependence of solutions on initial conditions, known as the butterfly effect, and the latter contains numerical sensitivities due to an ill-conditioned matrix with a large condition number (i.e., a large variance of growth rates). Here, we illustrate that the existence of a saddle point at the origin is a common feature that produces instability in both systems. Within the chaotic regime of the L63 nonlinear model, unstable growth is constrained by nonlinearity, as well as dissipation, yielding time varying growth rates along an orbit, and, thus, a dependence of (finite) predictability on initial conditions. Within the L69 linear model, multiple unstable modes at various growth rates appear, and the growth of a specific unstable mode (i.e., the most unstable mode during a finite time interval) is constrained by imposing a saturation assumption, thereby yielding a time varying system growth rate. Both models were interchangeably applied for qualitatively revealing the nature of finite predictability in weather and climate. However, only single type solutions were examined (i.e., chaotic and linearly unstable solutions for the L63 and L69 models, respectively), and the L69 system is ill-conditioned and easily captures numerical instability. Thus, an estimate of the predictability limit using either of the above models, with or without additional assumptions (e.g., saturation), should be interpreted with caution and should not be generalized as an upper limit for atmospheric predictability.

**Keywords:** Lorenz model; chaos; instability; saddle point; SDIC; sensitivities; finite predictability; ill-conditioned

## 1. Introduction

Two pioneering studies by Dr. Lorenz [1,2] changed our view on the predictability of weather, and turned our attention from regularity and unlimited predictability associated with Laplace's view of determinism to the irregularity and finite predictability associated with Lorenz's view of deterministic chaos. Chaos is defined as the sensitive dependence of solutions on initial conditions (SDIC), known as the butterfly effect [3]. The feature of SDIC reveals the difficulty in obtaining accurate, long-term predictions, suggesting a finite predictability (e.g., [4]).

Over the past several years, pioneering, yet incomprehensive, theoretical results derived from a limited collection of the above, and other studies, have led to an increased understanding (or misconceptions, to be illustrated) that the conventional view of "weather is chaotic" and that the so-called theoretical limit of predictability of two weeks are well supported by Lorenz's 1963 (L63) and 1969 (L69) studies [1,5]. As a result, when real-world global models produced encouraging simulations at extended-range (15–30 day) time scales [6–9], people who believe in the predictability limit of two weeks have interpreted these new results as inconsistent with chaos theory (e.g., [10]). While some researchers have applied nonlinear L63-type models for understanding the chaotic nature of weather and climate, other researchers have applied the major findings of linear L69-type models for estimating a predictability horizon for the atmosphere. Over the past several decades, researchers in the field of nonlinear dynamics have continuously improved our understanding of nonlinear responses, as well as local and global stability, within Lorenz-type models. Some recent theoretical studies may potentially provide justifications for promising extended-range simulations. However, due to ineffectiveness and difficulties in exchanging ideas and sharing results in different disciplines, related findings are, unfortunately, not fully known within the Earth science community. In this study, dynamical systems methods that are now fairly standard in recent nonlinear studies are applied in order to reveal the unreported features of the classical Lorenz model(s) (in particular, the L69 model).

Based on a comprehensive literature review, we believe current barriers to the advancement of weather and short-term climate predictions originate from gaps between the "improved understanding" of predictability with multistability, derived from advanced theoretical models, and the current approach, based on the conventional, yet incomplete, understanding of predictability with only SDIC and monostability. In contrast to the monostability that allows single type solutions (i.e., chaotic solutions in [1] and unstable solutions in [5]), the concept of multistability that contains coexisting chaotic and non-chaotic solutions has been emphasized (e.g., [11–14]). Recently, based on the concept of time varying multistability, refs. [15,16] provided a revised view that "weather possesses chaos and order; it includes emerging organized systems (such as tornadoes) and recurrent seasons". Such a revised view that suggests distinct predictability for chaotic and non-chaotic systems may lay a foundation for a potential predictability beyond Lorenz's predictability limit of two weeks. The revised view was proposed based on the classical and generalized L63 models that are well studied in nonlinear dynamics (e.g., [11–14,17–31]).

By comparison, it is the L69 model that has mainly been applied for addressing predictability in meteorology (e.g., [32,33]). Since statistical methods were applied to derive the L69 model, which is not a turbulence model but laid a foundation for the development of turbulence models (e.g., [34,35]), the L69 model has been analyzed using methods different from those of dynamical systems. On the other hand, the advantages of applying dynamical systems methods for investigating turbulence models have been documented (e.g., [36,37]). Although eigenvalue problems associated with L69-type models have previously been solved, only features for the largest eigenvalues have been documented. From the perspective of dynamical systems, an analysis of both positive and negative eigenvalues, as well as the corresponding exponential and oscillatory solutions, is helpful for understanding predictability within a multiscale system. Thus, by applying a unified, although simple, approach in a reanalysis of the models, fundamental questions can be revisited, including: (1) what type of mathematical, as well as physical, relationship exists between the two models and (2) what are their similarities and difference in spatial and temporal scale interactions. We hope that such an analysis will help researchers who are familiar with one type of model to quickly capture the major features and findings of the other type of model.

In this study, we perform new analyses, together with a literature review, to provide insights on the L63 and L69 models in terms of two types of sensitivities and the common feature of a saddle point. We then discuss how specific features of the two models were previously applied to determine finite predictability. Section 2 reviews general features of the L63 and L69 models. Similarities and differences of the two models are presented

in Section 3. Concluding remarks are provided at the end. Appendix A discusses the full L63 model and its simplified versions, including the non-dissipative L63 model [29]. Part I of Supplementary Materials includes the following two sections: Section (A): a simple illustration of ill-conditioning, and Section (B): an Illustration of a stiff ODE. Part II of Supplementary Materials summarizes additional features of the L63 model regarding SDIC and finite predictability.

## 2. The Lorenz 1963 and 1969 Models

In this section, through new analyses and a literature review, we discuss general features of the L63 and L69 models in order to propose a simple, 2nd-order ODE that retains the common properties of the two models. Based on the proposed 2nd-order ODE, specific features of the L63 and L69 models, including two types of sensitivities and the impact of a saddle point, are discussed in Section 3.

### 2.1. The L63 Limited-Scale, Nonlinear Model

The L63 model has been discussed in numerous studies [17,19–21], including in our studies [11,12,25–31] and references therein. A brief summary of the model is provided here, along with mathematical equations for the full and simplified versions found in Appendix A. Based on a system of partial differential equations that describe the time evolution of vorticity and temperature for the Rayleigh–Benard convection [38]), a system of three, 1st-order ODEs (e.g., Equations (A1)–(A3)) were obtained for rediscovering the SDIC [1]. Three major physical processes are heating, dissipation, and nonlinear processes. Such a model is called the Lorenz 1963 (L63) model.

As discussed in Appendix A, the system of three, 1st-order ODEs can be transformed into a system containing one 2nd-order and one 1st-order ODEs (e.g., Equations (A6) and (A7); also see [31]). Such a system with (or without) dissipations can easily be compared to the Pedlosky model [39–43] (or the Duffing Equation) to reveal mathematical universality amongst the systems (e.g., [30,31]). By ignoring dissipative terms, the 1st- and 2nd-order ODEs become uncoupled. The uncoupled, 2nd-order ODE is referred to as the non-dissipative L63 model (e.g., Equation (A11)), which is written as:

$$\frac{\mathrm{d}^2 X}{\mathrm{d}\tau^2} - \lambda^2 X + \frac{X^3}{2} = 0. \tag{1}$$

Here, $\tau$ is dimensionless time. The constant $\lambda^2$ is proportional to the product of $\sigma$ and $r$ that represent the Prandtl number and the normalized Rayleigh number, respectively. In general, $\lambda^2$ is also a function of initial conditions and, thus, $\lambda^2 = \sigma r + constant$.

Without a loss of generality, $\lambda^2$ is assumed to be positive (i.e, $\lambda^2 = \sigma r > 0$) within the non-dissipative L63 model. Thus, the linear version of the system produces both stable and unstable modes (i.e., $e^{-\gamma t}$, $e^{\gamma t}$, and $\gamma = \sqrt{\lambda^2}$). The role of nonlinearity is discussed in Section 3.1.

To facilitate discussions below, the energy function of the above system is obtained by multiplying both sides of Equation (1) by $dX/d\tau$ and by performing an integration with respect to $\tau$ (e.g., [23,31]):

$$TE = \left[\frac{1}{2}\left(\frac{dX}{d\tau}\right)^2\right] + \left[-\frac{1}{2}\lambda^2 X^2 + \frac{1}{8}X^4\right] = C. \tag{2a}$$

Here, $TE$ indicates the total energy. The first and second pairs of brackets represent the kinetic energy and potential energy, respectively (see [31] for details). $C$ is a constant that is determined by an initial condition. Within a linear system (i.e., no cubic term in Equation (1)), we have:

$$TE = \left[\frac{1}{2}\left(\frac{dX}{d\tau}\right)^2\right] + \left[-\frac{1}{2}\lambda^2 X^2\right] = C \quad \text{(for a linear system).} \tag{2b}$$

Later, to illustrate solution characteristics, we compute the contour lines of total energy for both linear and nonlinear systems. We specifically examine the special case of $TE = 0$ (i.e., $C = 0$).

### 2.2. The L69 Multiscale, Linear Model

By applying a modified, quasi-normal (QN) closure which assumes the 4th cumulants are zero (e.g., [44]; i.e., relating the fourth-order velocity correlations to second-order velocity correlations), Lorenz proposed the L69 model based on the following two-dimensional, conservative vorticity equation:

$$\frac{d\nabla^2\Psi}{d\tau} = 0, \tag{3}$$

where $\Psi$ and $\nabla^2\Psi$ represent the $2D$ $(x, y)$ stream function and the vorticity, respectively.

Such an approach was extended to yield the Eddy-Damped QN(EDQN) approximation, that replaces the 4th cumulants by a linear damping term, and then the EDQN Markovian (EDQNM) approximation using a minor modification called the Markovianization [34,44,45]. While Leith, 1971 [34] applied a model with the EDQNM approximation in order to study the predictability of 2D turbulence, ref. [35] proposed the test-field mode, overcoming the issue using an artificial cutoff in nonlinear interactions in the Leith 1971 model, for determining the predictability of turbulent flows. In regards to the above models [5,34,35], a common assumption was that: *"Estimates of the predictability of the planetary-scale motions of the atmosphere have been based on turbulence models in which the atmosphere is treated as an isotropic homogeneous two-dimensional turbulent fluid."* [35].

Since many Fourier modes were used for derivations, the L69 model contains multiple, physical modes. While used to illustrate the statistics of predictability within a multiscale framework, the model is not a turbulence model as compared to the models of Leith, 1971 and Leith and Kraichan, 1972 [34,35].

For revealing major features of solutions, linearization was applied to yield the following model in matrix form (e.g., Equation (43) of Lorenz, 1969 [5]):

$$\frac{d^2\overrightarrow{W}}{d\tau^2} = \mathbf{A}\overrightarrow{W}. \tag{4}$$

Here, $\mathbf{A}$ is a $N \times N$ time-independent matrix, and $N$ is the total number of wave modes. Each element within the matrix $\mathbf{A}$ represents the scale interactions of two Fourier modes (Equations (2a) and (2b) of Durran and Gingrich, 2014 [33]). $\overrightarrow{W}$ represents a column vector consisting of $N$ state variables $W_k$, $k = 1, 2, \cdots, N$. $k$ is the wavenumber, and $W_k$ represents the ensemble mean of the kinetic energy of the perturbations for the wave mode $k$. Equation (4) indicates that the L69 model is physically multiscale with $k = 1, 2, \cdots, N$, and also mathematically linear. For a comparison to Equation (2b), the energy function for Equation (4), which contains a symmetric matrix, is written as follows:

$$TE = \left[\frac{1}{2}\left(\frac{d\overrightarrow{W}}{d\tau}\right)^2\right] + \left[-\frac{1}{2}\overrightarrow{W^T}\mathbf{A}\overrightarrow{W}\right] = C.$$

Here, $\overrightarrow{W^T}$ is the transpose of the column vector $\overrightarrow{W}$. The above equation becomes (2b) when $N = 1$. In general, since $A$ is not a symmetric matrix, the so-called similarity transformation is applied for diagonalization to obtain the above equation. Thus, the vector $\overrightarrow{W}$ is replaced by the corresponding transformed column vector.

Since the L69 model is linear, it is important to understand how the model can help reveal the features of the original system (i.e., a partial differential equation for the conservation of vorticity). From a dynamical system perspective, the foundation of linearization is rooted in the linearization theorem, also known as the Hartman–Grobman Theorem (e.g., [22,46]): Suppose the $N$-dimensional system has an equilibrium point at $X_c$ that is

hyperbolic (i.e., with non-zero growth or decay rates). If so, nonlinear flow is then conjugate (i.e., dynamically equivalent) to the flow of the linearized system in the neighborhood of $X_c$. Within the L69 model, the origin is a saddle point and, thus, a hyperbolic point. As a result, the L69 model may describe the solution of the corresponding nonlinear system near the origin. Due to the limit of a linear system, a saturation assumption was imposed to prevent the unbounded growth of perturbations, as follows: when an unstable mode grows to reach its maximum, as determined by the selected spectrum, it is removed from the system (e.g., [32,47]. Then, a predictability horizon for a specific unstable mode represents the time when the mode becomes saturated. Thus, the impact of the saturation assumption on the estimate of predictability should be examined, and is discussed in Section 3.

From a dynamical perspective, Equation (4) can be transformed into a system of $2N$, 1st order ODEs by introducing additional $N$ state variables that represent 1st-order time derivatives of the original $N$ state variables. The $2N$ state variables can be used as coordinates for constructing the so-called phase space for analysis, as discussed in Section 3. Critical points are defined when the right hand side of the system of $2N$ ODEs becomes zero.

## 3. Discussions

In the discussion, we first analyze the saddle point and instability within a simple, linear, 2nd-order ODE, then turn the system into the non-dissipative L63 model by including a cubic term, and then present periodicity and limit chaos within the non-dissipative L63 model. We then illustrate specific features of the L69 model, including a saddle point, numerical instability, and numerical sensitivities; and provide comments on a L69-based conceptual model for a chain process. Based on the SDIC of the L63 model and the multiscale instability of the L69 model, we also discuss how to estimate predictability.

### 3.1. Features of the L63 Model
#### 3.1.1. Physical vs. Numerical Instability within a Linear 2nd-Order ODE

Based on Equations (1) and (4), which represent the L63 and L69 models, respectively, a simple linear, 2nd-order ODE is proposed that is, in reality, Equation (1) without the cubic term, as follows:

$$\frac{\mathrm{d}^2 X}{\mathrm{d}\tau^2} = \lambda^2 X. \tag{5}$$

$\lambda^2$ is positive within the L63 model and is either positive or negative within the L69 model. A system that contains a positive value for $\lambda^2$ possesses both stable an unstable modes (i.e., $e^{-\gamma\tau}$ and $e^{\gamma\tau}$ for a positive $\gamma = \sqrt{\lambda^2}$). A general solution can be expressed as the linear combination of these two modes (i.e., $c_1 e^{-\gamma\tau} + c_2 e^{\gamma\tau}$, $c_1$ and $c_2$ are determined by initial conditions). As a result, given the same model parameter, solutions may contain a stable mode, an unstable mode, or both, depending on the initial condition.

While an unstable mode may be physically interesting and important (e.g., [48]), in the real world, a stable mode likely represents a more realistic solution in response to initial small-scale perturbations. For example, given a tiny perturbation (e.g., any human's flap) as an initial condition, we may select a stable solution if the corresponding response can be described by Equation (5). However, as illustrated by an example that produces an analytical, stable solution [49], an unstable mode may be incorrectly captured by a numerical method. Such a feature, referred to as numerical instability, is reviewed below using Equation (5).

Given $\lambda^2 = 10\pi^2$ ($\sim$98.7) and an initial condition of $X(0) = 1$ and $X'(0) = -\sqrt{10}\pi$, an analytical, stable solution of $X = e^{-\sqrt{10}\pi\tau}$ can be obtained. Such an initial condition yields $TE = 0$ in Equation (2b) for the linear system. Values for the solution that is a monotonically decreasing function of time are listed in the 3rd column of Table 8.6.4 of Boyce and DePrima, 2012 [49]. By comparison, when the Runge–Kutta scheme is applied, numerical solutions within the 2nd column of their Table increase with time, indicating an instability. A detailed calculation (not shown) indicates that the growth rate of the numerical, unstable solution

is $\sqrt{10}\pi$. Therefore, the occurrence of such an instability appears because: (1) given the same parameters, the model (Equation (5)) allows both stable and unstable modes, and numerical errors produce a tiny, but non-zero, coefficient for the unstable mode; and (2) the error amplifies at the growth rate of the unstable mode and quickly dominates due to a large growth rate (i.e., $\sqrt{10}\pi$). Below, we analyze the solution within a 2D phase space for comparison to a nonlinear solution, and then illustrate that such numerical instability can easily be captured within the L69 model.

### 3.1.2. A Perspective of Dynamical Systems: Phase Space and a Saddle Point

From the perspective of dynamical systems, the above feature can be qualitatively illustrated using the following system of 1st-order ODEs derived from Equation (5):

$$\frac{dX}{d\tau} = Y, \tag{6a}$$

$$\frac{dY}{d\tau} = \lambda^2 X. \tag{6b}$$

A new variable $Y = X'$ is introduced. Using $X$ and $Y$ (i.e., $X$ and $X'$) as coordinates, a two-dimensional phase space can be constructed. As previously discussed, a solution with a positive value of $\lambda^2$ is written as:

$$X = c_1 e^{-\gamma\tau} + c_2 e^{\gamma\tau}, \tag{7a}$$

yielding:

$$\begin{pmatrix} X \\ Y \end{pmatrix} = c_1 e^{-\gamma\tau} \begin{pmatrix} 1 \\ -\gamma \end{pmatrix} + c_2 e^{\gamma\tau} \begin{pmatrix} 1 \\ \gamma \end{pmatrix}. \tag{7b}$$

Here, $Y = X'$ and $\gamma = \sqrt{\lambda^2} = \sqrt{10}\pi$. The first and second vectors on the right hand side of the equation are referred to as the stable and unstable eigenvectors, respectively, within a phase space. Thus, when an orbit begins near the origin $(X, Y) = (0,0)$, it may move away from the origin in the direction of the unstable eigenvector or approach the origin in the direction of the stable eigenvector. The origin $(X, Y) = (0,0)$ is called a saddle point. Stable and unstable eigenvectors within a linear system can be generalized to become stable and unstable manifolds of a saddle point within a nonlinear system, respectively: the stable (unstable) eigenvector is tangent to the stable (unstable) manifold at the saddle point (e.g., [50]).

Based on the total energy in Equation (2b), which describes the relationship between $X$ and $Y$ (i.e., $X'$), a solution trajectory can be analyzed using the contour lines of total energy. As shown in Figure 1, two straight lines with $TE = C = 0$ (i.e., $X' = \sqrt{10}\pi X$ and $X' = -\sqrt{10}\pi X$) indicate the unstable and unstable eigenvectors, respectively. In the previous example, the stable solution, starting at the initial condition of $(X, Y) = (1, -\sqrt{10}\pi)$, moves towards the origin along the zero contour line (Figure 1a,b). Ideally, the orbit should eventually arrive at the origin and stay there forever. However, since the origin is a saddle point, it is numerically challenging to simulate that type of evolution. When the orbit moves very close to the origin where the value of both $X$ and $X'$ are small, any tiny noise (caused by round-off errors) may lead the orbit to move away from the saddle along the direction of the unstable eigenvector (i.e., the line of $X' = \sqrt{10}\pi X$). A similar challenge within a nonlinear system, as documented in Shen, 2020 [30], is reviewed in the next subsection.

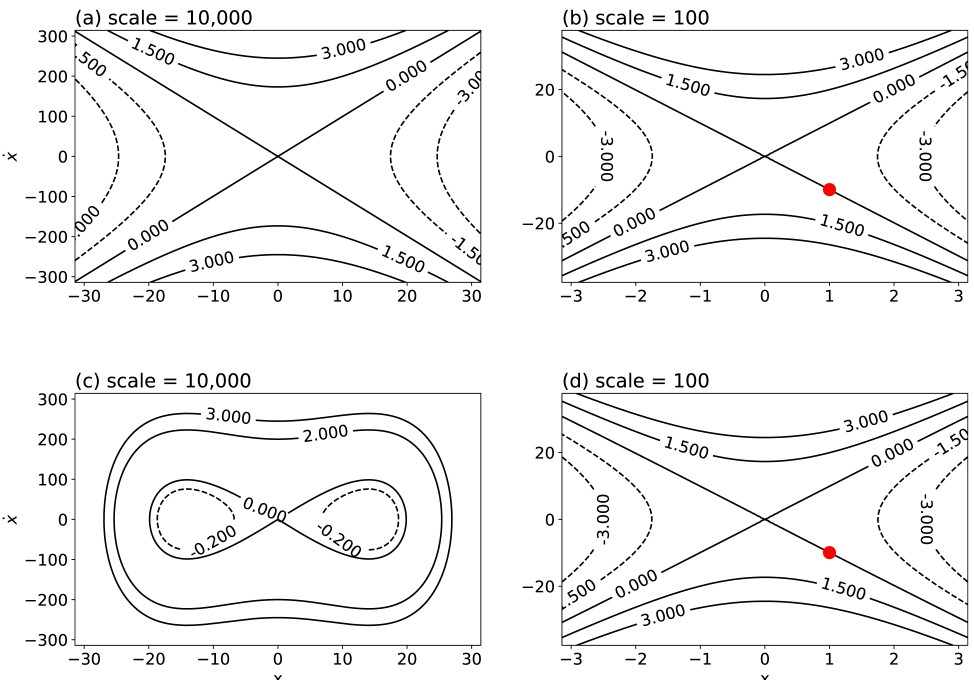

**Figure 1.** The contour lines of total energy in Equations (2a) and (2b) for the linear (**a**,**b**) and nonlinear (**c**,**d**) systems of $X'' - 10\pi^2 X + X^3/2 = 0$. Panels (**b**,**d**) provide a zoomed-in view of panels (**a**,**c**), respectively. A large red dot at $(X, X') = (1, -\sqrt{10}\pi)$ displays a starting point of an orbit that moves along the zero contour line towards the origin.

By comparison, a negative eigenvalue ($\lambda^2 < 0$) produces a general periodic solution with trigonometric functions of *sin* and *cos*. The corresponding solution displays a closed curve within the 2D $X - X'$ phase space, and the critical point $(0, 0)$ is called a center (Table 1).

**Table 1.** The classification of various types of solutions for a linear 2D system with real coefficients. We assumed an arbitrary $\alpha$, $\beta > 0$, and $\gamma > 0$.

| Characteristics | Solutions | Critical Points | Remarks |
|---|---|---|---|
| non-oscillatory | $c_1 e^{\gamma\tau} + c_2 e^{-\gamma\tau}$ | saddle | monotonic |
| oscillatory ($\beta \neq 0$) | $c_1 e^{(\alpha+i\beta)\tau} + c_2 e^{(\alpha-i\beta)\tau}$ | | |
| | $\alpha = 0$ | center | periodic |
| | $\alpha > 0$ | spiral source | |
| | $\alpha < 0$ | spiral sink | |

### 3.1.3. Periodicity and Centers Enabled by Nonlinearity

Mathematically, it is natural to add nonlinear terms, $X^2$ or $X^3$, into the linear system in Equation (5) in order to constrain growth of the unstable mode. As previously illustrated, the inclusion of the quadratic and cubic terms yields systems comparable to the Korteweg–de Vries (KdV) and nonlinear Schrodinger equations, respectively, in a traveling-wave coordinate [30,51–55]. The latter can be mathematically written as the non-dissipative L63 model in Equation (1), as follows:

$$\frac{\mathrm{d}^2 X}{\mathrm{d}\tau^2} - \lambda^2 X + \frac{X^3}{2} = 0,$$

which also represents an unforced Duffing Equation (e.g., [29,56]).

Inclusion of the nonlinear cubic term leads to two, non-trivial critical points at $X = \pm 2\lambda$ and $\lambda > 0$ (e.g., [31]). To reveal the role of the nonlinear term, we can simply compare the contour lines of the energy functions, Equations (2a) and (2b), for the nonlinear and linear systems, Equations (1) and (5), respectively. As shown in Figure 1c, two centers appear, as indicated by a family of nearby closed contour lines that enclose one of the centers. Each of the closed curves represents one periodic solution with a specific initial condition.

As a result of the two centers and the saddle point at the origin, small- and large-cycle oscillations, and homoclinic orbits appear for $C < 0$, $C > 0$, and $C = 0$ in Equations (2a) and (2b), respectively. As shown in Figure 1c, for $C \neq 0$, each closed contour line indicates an oscillatory solution (see [29,30,57,58]). For $C = 0$, an orbit may move in the direction of the unstable manifold of a saddle point and return back along the direction of the stable manifold of the same saddle point. Such an orbit connecting the unstable and stable manifolds of the same saddle point is called a homoclinic orbit. The number "8" shape in Figure 1c indicates two homoclinic orbits. Thus, all three types of solutions reveal how nonlinearity limits the growth of the unstable solution, producing bounded solutions and global stability.

On the other hand, while an unstable eigenvector can be easily captured using a numerical method even though the initial conditions only allow solutions with a stable eigenvector in a linear system, such a feature also appears in association with the homoclinic orbit within a nonlinear system. Below, an analysis of the homoclinic orbit in Shen, 2020 [30] is briefly reviewed for a comparison.

### 3.1.4. (Computational) Limit Chaos Associated with a Homoclinic Orbit

The association of a saddle point with SDIC has been illustrated using the so-called linear geometric model [59] and the nonlinear non-dissipative L63 model [11,29]. Although the two simplified models may not exactly produce the same type of SDIC as the full L63 model, they may serve as a baseline system for revealing solution sensitivities. Sensitivities associated with the special homoclinic orbit were previously documented in Figure 4 of Shen, 2020 [30]. Most solutions within the non-dissipative L63 model are oscillatory and only two of them are homoclinic orbits. As a result, such sensitivities may be referred to as limit chaos [3]. Since it is challenging to avoid limit chaos in numerical integrations of the non-dissipative L63 model, we may simply call the sensitivity computational limit chaos.

### 3.1.5. Spiral Sinks Associated with an Additional Dissipative Term $(-\epsilon Y)$

In general, saturation of an unstable mode requires the existence of a stable critical point. However, the above discussions suggest that the appearance of two centers cannot enable unstable modes to become saturated (i.e., the solution reaches a constant). To obtain non-trivial stable critical points that can help yield steady-state solutions, one dissipative term $(-\epsilon Y)$ is added into the non-dissipative L63 model (e.g., Equation (A2)), yielding:

$$\frac{\mathrm{d}^2 X}{\mathrm{d}\tau^2} + \epsilon \frac{dX}{d\tau} - \lambda^2 X + \frac{X^3}{2} = 0, \tag{8}$$

which can also be obtained from Equation (A10) that represents a simplified Lorenz model with only one dissipative term $(-Y)$ (i.e., without two dissipative terms, $-\sigma X$ and $-bZ$).

Following the derivations for Equations (2a) and (2b), the energy function for Equation (8) indicates that the system energy decreases with time, producing steady state solutions. This is consistent with the analysis of critical points, showing two, stable, non-trivial critical points as spiral sinks [31]. A mathematical solution for a general spiral sink is listed in Table 1. As summarized in Table 3 of Shen, 2021 [31], the locations of the critical points for cases $\epsilon = 0$ and $\epsilon \neq 0$ in Equation (8) are the same. Therefore, it is relatively easy to make a comparison. While the model for $\epsilon = 0$ displays regular oscillatory solutions (e.g., Figure 7 of Shen, 2018 [29]), the model for $\epsilon \neq 0$ simulates steady-state solutions, as shown in Figure 2. For steady-state solutions that indicate a saturation, an orbit moves

from the saddle point and towards one spiral sink. However, the time evolution of the solution is not a monotonic function in time.

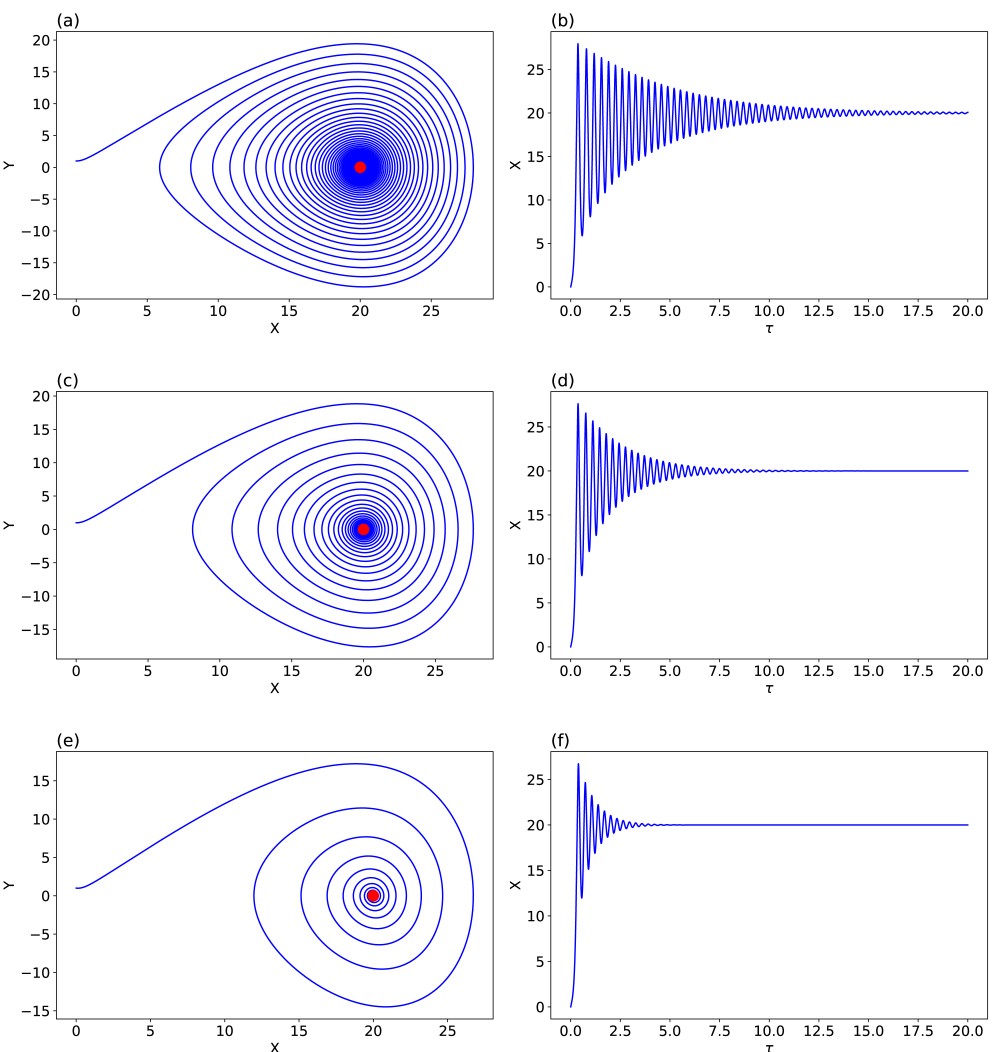

**Figure 2.** Solutions for the simplified Lorenz model that only contains one dissipative term, $-\epsilon Y$, and $\epsilon = 0.5, 1$, and 2.5, from top to bottom. Left panels (**a**,**c**,**e**) display orbits within the $X - Y$ space, while right panels (**b**,**d**,**f**) depict the time evolution of $X$. A stable non-trivial critical point is indicated by a large red dot.

### 3.1.6. SDIC and Finite Predictability Within the L63 Model

The full L63 model has been widely applied for illustrating the SDIC, as follows. Two runs, the control and parallel runs, are performed using the same model and the same parameters, but slightly different initial conditions (ICs). For example, a tiny perturbation of $\epsilon = 10^{-10}$ is added into the initial condition for the parallel run. Then, the time evolution for the two solutions is analyzed. As a result of the tiny perturbations, the two runs almost produce identical results during an initial period of time. As shown in Figure 3, this feature is called continuous dependence on ICs [50]. However, in spite of the initial tiny perturbations, two runs produce very different solutions after a certain period of time. Such a feature with very different results is referred to as SDIC. Similar results have been documented within the scientific literature (e.g., Figure 2 of Shen, 2019b [8] and Figure 1 of Shen et al., 2021b [16]). Combined CDIC and SDIC determine the predictability horizon. Namely, the predictability limit is determined before the onset of SDIC, and the predictability horizon displays a dependence on initial conditions that are close to or away from one of three critical points within the L63 model [15,60–62].

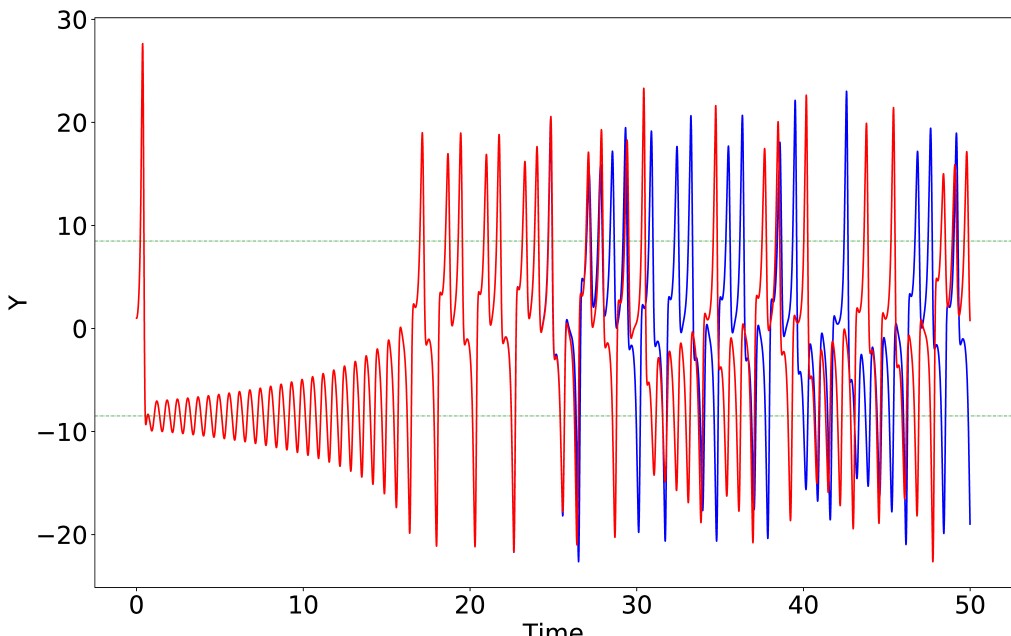

**Figure 3.** Two nearby trajectories within the L63 model with $r = 28$ and $\sigma = 10$. Solutions, in different colors, were obtained from the control and parallel runs. The only difference between the two solutions is that the parallel run adds a small perturbation ($1 \times 10^{-10}$) into the initial value of Y. The sensitive dependence of solutions on the initial conditions is indicated by the divergence of two orbits in blue and red. The two horizontal lines indicate the location of the non-trivial critical points.

The above discussions qualitatively reveal the finite predictability of the L63 model. Within the nonlinear L63 model, its chaotic feature is indicated by the appearance of one positive Lyapunov exponent (e.g., [25,26] and references therein). The Lyapunov exponent (LE) is defined as the long-term average exponential rate of divergence of nearby trajectories [63–65] (i.e., the long-term average growth rate of an infinitesimal error [66]). The LE is calculated by evaluating the derivative along the direction of maximum expansion and averaging its logarithm along the trajectory (e.g., [67,68]). Although earlier studies made an attempt to apply the largest Lyapunov exponent for estimating predictability, a general average predictability is less interesting as compared to short-term behavior [66]. Additionally, the L63 limited-scale model with one positive LE is too simplified to estimate predictability in weather and climate. On the other hand, a challenge in extending the predictability horizon by improving the accuracy of initial conditions can be illustrated by assuming the error (i.e., separations of nearby trajectories) to be proportional to $e^{L\tau}$, here $L$ and $\tau$ represent a LE and time, respectively (see [23] for details). However, such a rough formula of $e^{L\tau}$ that does not take SDIC into consideration should be applied with caution.

As discussed, when using the linear and nonlinear versions of the non-dissipative L63 model (e.g., Equation (1)), it is challenging to simulate the evolution of an orbit that moves towards the saddle point in a two-dimensional or higher phase space. By comparing the geometric and Lorenz models, such a feature is shown to be related to SDIC.

### 3.2. Features of the L69 Model

Based on the L63 model, the previous discussions illustrate the different roles of nonlinearity, which are missing in the L69 model. In comparison, within or based on the L69 model, the following features are discussed: (1) eigenvalues and eigenvectors; (2) a conceptual model for a chain process; (3) numerical instability associated with large eigenvalues; (4) ill-conditioning associated with large condition numbers; and (5) monotonicity within the L69 model. Based on the L69 and L63 models, we then provide comments on the estimate of predictability.

### 3.2.1. Eigenvalues and Eigenvectors

As shown in Equation (4) of this study or Equation 43 of Lorenz, 1969 [5], the L69 model consists of a linear system of $N$, 2nd-order ODEs, here $N = 21$. Such a system may be transformed to form an eigenvalue problem for an analysis. By assuming a solution vector in the form of $\vec{W} = e^{\lambda \tau} \vec{V}$, we can convert Equation (4) into the following equation:

$$\mathbf{A}\vec{V} = \lambda^2 \vec{V}. \tag{9}$$

Here, $\vec{V}$ and $\lambda^2$ represent the eigenvector and eigenvalue of the matrix $\mathbf{A}$, respectively. This approach is made possible because Equation (4) does not contain 1st-order derivatives. Here, $\lambda^2$, instead of $\lambda$, represents an eigenvalue. A system of $N$, 2nd-order ODEs produces $N$ eigenvalues using this approach. As compared to Equation (5) that contains "one" eigenvalue, each of the $N$ eigenvalues in Equation (9) is associated with two components, $e^{\gamma \tau}$ and $e^{-\gamma \tau}$ for $\gamma = \lambda^2$. Details are provided below.

To simplify discussions, we consider distinct eigenvalues that can be positive or negative. Positive and negative eigenvalues yield real and pure imaginary values for $\gamma$, representing exponential modes and oscillatory modes, respectively. Here, we compute the eigenvalues and eigenvectors for the six systems available from Rotunno and Synder, 2008 and Durran and Gingrich, 2014 [32,33]. For these systems, only $9 \times 9$ matrices were listed in the studies. However, as discussed below, this limitation does not have a significant impact on our major conclusion since the $9 \times 9$ matrices represent sub-matrices of the original $21 \times 21$ matrices at larger scales. Therefore, the eigenvalues are relatively small as compared to the eigenvalues of the corresponding full matrices. References to the six matrices are provided in Table 2.

**Table 2.** Condition numbers in six L69-type systems from Rotunno and Synder (2008, RS08) and Durran and Gingrich (2014, DG14) [32,33]. Both Python and Matlab were used for calculations and displayed similiar results.

|  | Python | Matlab | Remarks |
|---|---|---|---|
| Table 1 of RS08 | $8.319352 \times 10^5$ | $8.3194 \times 10^5$ | 2DV dynamics |
| Table 2 of RS08 | $8.446532 \times 10^5$ | $8.4465 \times 10^5$ | vs. Lorenz (1969) |
| Table 3 of RS08 | $2.791518 \times 10^4$ | $2.7915 \times 10^4$ | "unlimited predictability" |
| Table 4 of RS08 | $2.146269 \times 10^9$ | $2.1463 \times 10^9$ | SQG dynamics |
| Table A1 of DG14 | $7.967004 \times 10^5$ | $7.9670 \times 10^5$ | vs. Table 1 of RS08 |
| Table A2 of DG14 * | $9.767672 \times 10^6$ | $9.7677 \times 10^6$ | vs. Table 4 of RS08 |

* There may be a typo in $C_{1,2}$ in Table A2 of DG14. A revised condition # is $O(10^9)$.

Figure 4 displays the nine eigenvalues of the matrix from Table 4 of Rotunno and Synder, 2008 [32]. Here, we observe positive and negative eigenvalues, both of which are large in magnitude. The first three positive eigenvalues are 457.0, 3.42, and 0.000472, respectively. A positive eigenvalue produces a pair of stable and unstable components, while a negative eigenvalue yields oscillatory components. Additionally, the system also produces eigenvalues with very small magnitudes. Due to the limit of finite precision, here, we do not make an attempt to determine whether (or not) these eigenvalues are zeros or even double zeros. Instead, we focus on eigenvectors with large eigenvalues.

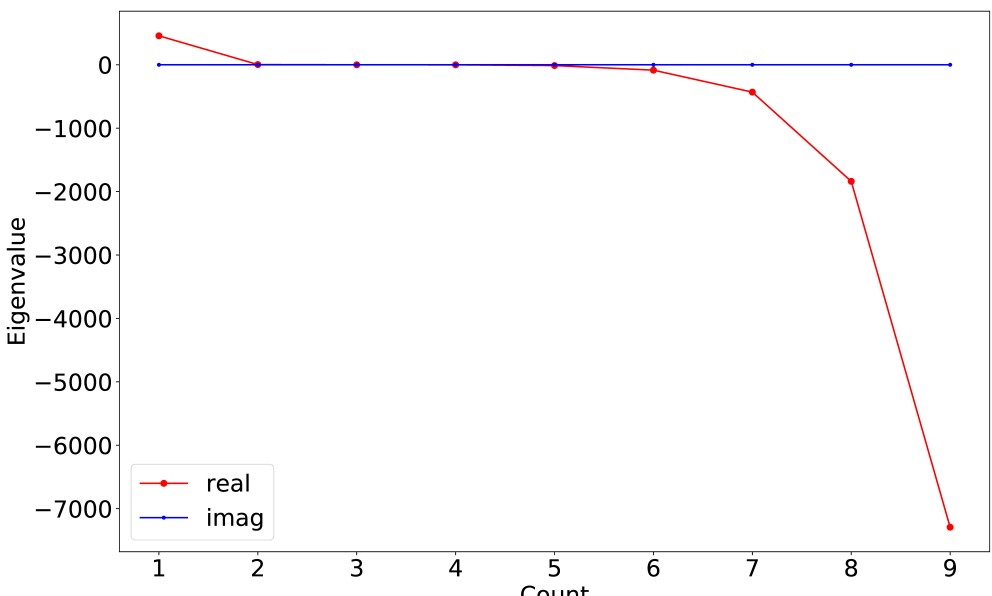

**Figure 4.** Nine eigenvalues for the $9 \times 9$ matrix from Table 4 of Rotunno and Synder, 2008 [32]. As discussed in the main text, each eigenvalue is associated with a pair of solution components. A positive eigenvalue yields a pair of exponentially growing and decaying components, while a negative eigenvalue leads to oscillatory components.

### 3.2.2. A Conceptual Model for a Chain Process

By analyzing the eigenvalues of the system in Equation (9), growth rates and predictability on various scales can be determined. Since the most unstable mode with the largest exponential growth rate should dominate, a system's predictability is solely determined by the largest growth rate (i.e., the largest $\sqrt{\lambda^2}$). On the other hand, the saturation assumption has been applied to extend a system's predictability as follows: when the unstable mode at the smallest scale reaches its saturated value, as determined by the energy spectrum, it is removed from the system. The time required for the most unstable component to become saturated defines a predictability horizon for the specific unstable mode. Afterwards, a subsystem with a sub-matrix (e.g., a $(N - 1) \times (N - 1)$ matrix) is obtained. Within the subsystem, different unstable modes appear, and the most unstable mode determines the predictability horizon. After becoming saturated, the mode is removed from the system. Thus, a system's predictability is obtained by adding all of the predictability horizons determined by each of the most unstable modes that sequentially appear, grow, and saturate. Simply speaking, a system's predictability is collectively determined by various eigenvalues at different scales. A mathematical analysis for an estimate of predictability is given near the end of this section. The above procedure relies on the successive eigenvalue calculation of matrices whose elements are continuously removed as a result of the saturation assumption (e.g., [32,47]). The procedure implies a conceptual model for a chain process that consists of repeated removal of the current most unstable mode and the appearance of a new most unstable mode.

The above conceptual model for the chain process is analyzed using Figure 5. Panels (a)–(f) display the first eigenvalues and the corresponding eigenvectors within a $9 \times 9$, $8 \times 8$, $7 \times 7$, $6 \times 6$, $5 \times 5$, and $4 \times 4$ matrix, respectively. Based on Figure 5, we can observe that the first eigenvector contains several non-zero components at smaller scales. Secondly, the first eigenvalue becomes smaller within smaller sub-matrices. Therefore, we may say that smaller scale modes possess larger growth rates. Since the first eigenvalue is always positive and since the corresponding eigenvector contains components at smaller scales, Figure 5 indicates the possibility for a chain process.

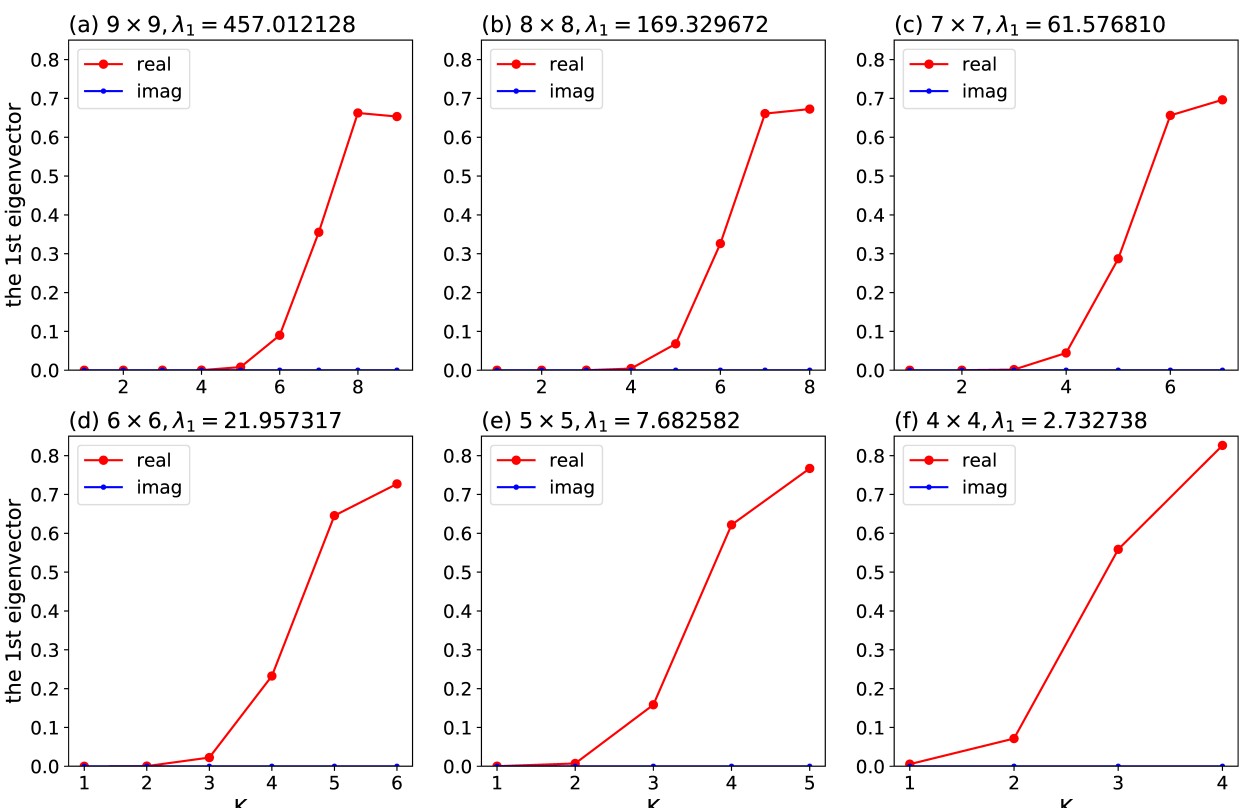

**Figure 5.** The first eigenvector of the $9 \times 9$ matrix from Table 4 of Rotunno and Synder, 2008 [32], and the $8 \times 8$, $7 \times 7$, $6 \times 6$, $5 \times 5$, and $4 \times 4$ sub-matrices, respectively.

While a specific eigenvector may contain more than one state variable, a specific state variable may appear in more than one eigenvector, depending on the ICs. Mathematically, we say that a general solution can be expressed in terms of a linear combination of all eigenvectors. As a result, a specific state variable (i.e., a specific wave mode) may concurrently grow at more than one growth rate. This feature complicates the process for estimating system predictability.

### 3.2.3. Numerical Instability Associated with Large Eigenvalues

Here, it should be noted that we cannot exclude the possibility for a specific state variable that appears as a decaying or an oscillatory component. Such a state variable should be more predictable. On the other hand, from a numerical perspective, such a solution with better theoretical predictability may be incorrectly simulated, appearing as an unstable mode. As discussed using Equation (5) in Section 3.1, it is easy for numerical methods to capture unstable modes even though a specific initial condition only allows a stable solution. In the example in Section 3.1, an eigenvalue of $\Lambda = \lambda^2 = 10\pi^2 \approx 98.7$ was used. As a result, numerical instability similar to that in Equation (5) may be easily found within the L69 system that possesses several large eigenvalues, including the largest eigenvalue $\Lambda_1 = 457.0$ within the $9 \times 9$ matrix and $\Lambda_1 = 169.3$ within the $8 \times 8$ matrix. Thus, it is reasonable to conclude that such numerical instability can be found within the $21 \times 21$ matrix.

On the other hand, as discussed in Section 3.1, nonlinearity may constrain the growth of unstable modes, and dissipation may reduce growth rates. Therefore, due to the lack of dissipation and nonlinearity within the L69 model, an estimate of predictability based on the growth rates of unstable modes should be interpreted with caution.

### 3.2.4. Ill-Conditioning Associated with Large Condition Numbers

In addition to potential numerical instability associated with large eigenvalues, here, we also reveal ill-conditioning with numerical sensitivities within the L69 model. Ill-conditioning is measured by a condition number ($\kappa(\mathbf{A})$), as follows:

$$\kappa(\mathbf{A}) = \|\mathbf{A}\|\|\mathbf{A}^{-1}\|, \tag{10}$$

where $\|\mathbf{A}\|$ represents the matrix norm of $\mathbf{A}$ and $\mathbf{A}^{-1}$ is the inverse matrix of $\mathbf{A}$. The above can be written as:

$$\kappa(\mathbf{A}) = |\frac{\Lambda_{max}}{\Lambda_{min}}|, \tag{11}$$

here, $\Lambda = \lambda^2$ represents an eigenvalue of matrix $\mathbf{A}$. Ill-conditioning occurs when the condition number ($\kappa(\mathbf{A})$) is large. Equation (11) suggests that a system with large variances in growth rates (i.e., containing a large ratio between the largest and smallest eigenvalues) is ill-conditioned. Such an expression is effective in revealing potential issues in sophisticated model systems that contain multiple scales and, thus, various growth rates. Within an ill-conditioned system, numerical results are sensitive to small changes, including round-off errors within the coefficient matrix. Sensitivity to small changes can be illustrated using a system of equations in Supplementary Materials modified from Kreyszig, 2011 [69], that contains a $2 \times 2$ matrix with a condition number of $20,001$. The system has a unique solution of $(0,0)$. However, when the system is perturbed by a tiny noise of $\delta$, its solution is shifted to $(5000.5\delta, 4999.5\delta)$ (e.g., $(0.50005, 0.49995)$ for $\delta = 10^{-4}$).

Table 2 displays large condition numbers for all of the six systems from Rotunno and Synder, 2008 and Durran and Gingrich, 2014 [32,33]. The condition numbers are on the order of $10^4$ or higher. As a result, we may expect large uncertainties in determining the location of a saddle point. On the other hand, from the perspective of transient solutions, a large condition number also indicates a large stiffness. As discussed using an example in Supplementary Materials, modified from [49] a system of stiff ODEs requires very small step sizes to obtain stable numerical solutions.

### 3.2.5. Solutions in Terms of Eigenvalues and Eigenvectors of the L69 Model

As a brief summary, the L69 model can be described as follows:

(1) The model is closure-based, physically multiscale, mathematically linear, and numerically ill-conditioned.
(2) The model possesses multiple positive and negative eigenvalues, and, thus, produces growing and decaying components and oscillatory components. However, the model may easily capture unstable modes due to numerical errors and large growth rates.
(3) Since the system is linear and homogeneous, the only equilibrium point is a trivial equilibrium (or critical) point at $\overrightarrow{W} = 0$. The critical point is a saddle point that contains multiple pairs of stable and unstable eigenvectors associated with multiple positive eigenvalues.

Mathematically, solutions of the L69 linear system can be obtained by performing numerical integration or using a linear combination of system eigenvectors and eigenvalues. Under the saturation assumption, a system's predictability is collectively determined by various eigenvalues at different scales. To illustrate this idea, we provide an analysis of the general solution using eigenvalues ($\gamma_k^2$) and eigenvectors ($\overrightarrow{V_k}$), as follows:

$$\overrightarrow{W} = C_1 e^{-\gamma_1 \tau} \overrightarrow{V_1} + D_1 e^{\gamma_1 \tau} \overrightarrow{V_1} + C_2 e^{-\gamma_2 \tau} \overrightarrow{V_2} + D_2 e^{\gamma_2 \tau} \overrightarrow{V_2} + \cdots + C_n e^{-\gamma_n \tau} \overrightarrow{V_n} + D_n e^{\gamma_n \tau} \overrightarrow{V_n}. \tag{12}$$

When $N = 1$, this leads to $\overrightarrow{W} = W_1$ and $\overrightarrow{V_1} = 1$, and Equation (12) yields:

$$\begin{pmatrix} W_1 \\ W_1' \end{pmatrix} = C_1 e^{-\gamma \tau} \begin{pmatrix} 1 \\ -\gamma \end{pmatrix} + D_1 e^{\gamma \tau} \begin{pmatrix} 1 \\ \gamma \end{pmatrix},$$

which is the same as Equation (7b). Here, we, once again, bring the reader's attention to the fact that the L69 model consists of $N$, 2nd-order ODEs and, thus, $2N$, 1st-order ODEs. When a saturation assumption is applied for determining the predictability horizon, an initial condition for each of the unstable modes is required.

3.2.6. Finite Predictability within the L69 Model

As discussed using Figure 3, the combined CDIC and SDIC suggest finite predictability within the L63 model. While SDIC is indicated by the divergence of two nearby trajectories that start moving towards different non-trivial critical points, two trajectories may closely move around one of the non-trivial critical points during the epoch of CDIC. In Figure 3, such is indicated by oscillatory solutions centered at the value shown in green between $\tau = 3$ and $\tau = 15$. A key message is that when two nonlinear solutions leave the saddle point and move towards one of the non-trivial critical points, both may still closely move and revisit the neighbors of the saddle point several times (e.g., between $\tau = 18$ and $\tau = 22$ in Figure 3) prior to their separation (e.g., after $\tau = 26$ in Figure 3). Revisiting the neighborhood of the saddle is not allowed within a linear system.

By comparison, the L69 multiscale model that has the advantage of including a realistic energy spectrum has been applied to estimate multiscale predictability in weather. On the other hand, due to the limit of linearity, two nearby unstable orbits within the L69 model monotonically diverge, which is different from time varying changes within the L63 model. Additionally, the monotonic increase of errors may not be applicable to oscillatory or decaying solutions.

In Equation (12), a specific state variable (i.e., a specific wave mode, $W_k$) may concurrently grow at more than one growth rate. This feature complicates the process for estimating system predictability. Below, to simplify discussions without the need for determining "initial conditions", the e-folding time is used in order to estimate the predictability horizon of the most unstable mode (e.g., [70]). The e-folding time is the time interval in which an exponentially growing quantity increases by a factor of $e$. Thus, given a specific mode $k$, with a dominant growth rate of $\gamma_k$, the predictability limit (or horizon) (i.e., an e-folding time) for the specific mode is inversely proportional to its growth rate, as follows:

$$T_k = \frac{1}{\gamma_k} \tag{13}.$$

Thus, a system's predictability horizon is proportional to the time required for systems at all scales to become saturated (i.e., to increase by a factor of $e$):

$$T \approx \frac{1}{\gamma_{9 \times 9}} + \frac{1}{\gamma_{8 \times 8}} + \frac{1}{\gamma_{7 \times 7}} \cdots \tag{14}$$

Namely, a system predictability horizon is determined by the sum of the reciprocals for all growth rates. In general, due to 21 modes within the L69 model, Equation (14) does not represent an infinite series.

We also may want to know the condition under which the predictability is finite. If Equation (14) is a geometric series with a factor of $1/2$, the sum is finite, as shown below:

$$T \approx 1 + \frac{1}{2} + \frac{1}{4} + \frac{1}{8} + \cdots = 2. \tag{15}$$

The above example with a common factor of $1/2$ is consistent with discussions of Palmer et al., 2014, Palmer, 2017 and Lorenz, 1969 [5,71,72] in regards to Lorenz's empirical formula:

*Except for the smallest scales retained, where the effect of omitting still smaller scales is noticeable, and the very largest scales, where $X_k$ does not conform to a $-2/3$ law, successive differences $t_k - t_{k+1}$ differ by a factor of about $2^{-2/3}$. If one chooses to*

*reevaluate $t_1$ by summing the terms of the sequence $t_1 - t_2, t_2 - t_3, \cdots,$ one is effectively summing a truncated geometric series.*

The above calculation applied several assumptions, including assumptions that: (1) the series in Equation (14) can be extended to have an infinite number of terms, and (2) the series contains a common factor of $2^{-2/3}$, even for different slopes of the kinetic energy spectrum. On the other hand, the following two series are not convergent:

$$\sum_{n=1} \frac{1}{n} = 1 + \frac{1}{2} + \frac{1}{3} + \frac{1}{4} + \cdots = \infty, \tag{16a}$$

$$\sum_{p\,prime} \frac{1}{p} = \frac{1}{2} + \frac{1}{3} + \frac{1}{5} + \frac{1}{7} + \cdots = \infty. \tag{16b}$$

The two series do not produce a finite number. Therefore, whether (or not) an extension of Equation (14) with an infinite number of terms produces a finite predictability is still a challenging question. On the other hand, since Equation (14) is based on several assumptions, its validity should be examined. Other than the above, the L69 model is not a turbulence model, and all weather systems cannot be turbulent forever. As a result, it may be legitimate to conclude that "practical predictability" within the L69 is finite. Here, the practical predictability indicates a dependence of predictability on mathematical formulas and ICs, in contrast to intrinsic predictability that only depends on a flow itself [73].

### 3.2.7. A Comparison of Monostability and Multistability

Given model parameters, the L63 model produces single type solutions, including steady-state, chaotic, and limit cycle solutions. Such a feature is referred to as monostability. By comparison, generalized Lorenz models possess coexisting chaotic and non-chaotic attractors that appear with the same model parameters but different initial conditions [11–14]. The feature of attractor coexistence is called multistability. Although the L69 model is linear, it does allow various types of solutions associated with positive or negative eigenvalues. Thus, the L69 model may be viewed to possess multistability, despite the fact that only unstable solutions have been a focus.

Based on the results with multistability, we recently suggested a revised view that weather possesses both chaos and order, in contrast to the conventional view of "weather is chaotic". Such a view is additionally supported by this study. While chaotic solutions of the L63 model or linearly unstable solutions of the L69 model produce finite predictability, non-chaotic regular solutions may have unlimited predictability (up to the lifetime of the system or the duration of forcing). Such a revised view that turns our attention from monostability to multistability is neither too optimistic nor pessimistic as compared to the Laplacian view of deterministic predictability and the Lorenz view of finite predictability.

### 4. Concluding Remarks

Both the Lorenz 1963 (L63) and 1969 (L69) models [1,5] have been applied in the past to illustrate finite predictability. An estimate of a predictability limit of two weeks was initially obtained using the L69 model. In this study, new analyses along with a literature review provide insights on the mathematical and physical relationship, two types of sensitivities, and the impact of a saddle point on the two types of sensitivities. The L63 and L69 models are derived from different partial differential equations. One system is for convection, and the other system is for the conservation of barotropic vorticity. The L63 model is limited-scale and nonlinear; and the L69 model is closure-based, physically multiscale, mathematically linear, and numerically ill-conditioned. The former possesses a sensitive dependence of solutions on initial conditions, known as the butterfly effect, and the latter contains numerical sensitivities resulting from an ill-conditioned matrix with a large condition number (i.e., a large variance of growth rates). A common feature that produces unstable components in both systems is the existence of a saddle point at the origin. A saddle point provides an essential ingredient for chaos within the L63 model and

for linear instability within the L69 model. Table 3 provides a summary for the L63 model, the geometric model, the non-dissipative L63 model, and the L69 model. All of the listed models reveal the common feature of a saddle point.

**Table 3.** The Lorenz 1963 (L63) model [1], the geometric model [59], the non-dissipative L63 model [29], and the Lorenz 1969 (L69) model [5]). The non-dissipative L63 model is derived from the L63 model without dissipative terms shown in red. For an extreme case of $N = 1$ and a positive eigenvalue, the L69 model is dynamically comparable to the non-dissipative L63 model without the cubic term shown in blue. A saddle point is a common feature amongst the four models.

| (1) The L63 Model | (2) The Geometric Model |
|---|---|
| $\frac{dX}{d\tau} = \sigma Y - \sigma X,$ | $\frac{dX}{d\tau} = -3X,$ |
| $\frac{dY}{d\tau} = -XZ + rX - Y,$ | $\frac{dY}{d\tau} = 2Y,$ |
| $\frac{dZ}{d\tau} = XY - bZ.$ | $\frac{dZ}{d\tau} = -Z.$ |
| **(3) The Non-dissipative L63 Model** | **(4) The L69 Model** |
| $\frac{d^2 X}{d\tau^2} = (\Omega_0 + \sigma r)X - \frac{X^3}{2},$ | $\frac{d^2 \vec{W}}{d\tau^2} = \mathbf{A}\vec{W},$ |
| $\Omega_0$: constant. | $\mathbf{A}$: $N \times N$ matrix, |
|  | $\vec{W}$: a vector for $N$ state variables. |

Within the chaotic regime of the L63 nonlinear model, unstable growth is constrained by nonlinearity, as well as dissipation, yielding bounded solutions and time varying growth rates along an orbit. The appearance of SDIC suggests a finite predictability that displays a dependence on initial conditions. Within unstable solutions of the L69 linear model, multiple growth rates at various scales exist. Unlimited growth of the most unstable mode is suppressed by artificially imposing the saturation assumption: a saturated unstable mode that reaches its upper limit within a finite interval is removed from the system, enabling the appearance of a new most unstable mode. Thus, a system's growth rate appears to be time varying and a system's predictability is collectively determined by various eigenvalues at different scales.

While both the L63 and L69 models suggest finite predictability, only single type solutions (e.g., chaotic solutions within the L63 model and linearly unstable solutions within the L69 model) were considered. The SDIC of the L63 model leads to finite predictability. By comparison, ill-conditioning and the appearance of numerical instability are likely responsible for the "finite predictability" of the L69 study, as suggested by the following quote from the L69 study: *two states of the system differing initially by a small "observational error" will evolve into two states differing as greatly as randomly chosen states of the system within a finite time interval, which cannot be lengthened by reducing the amplitude of the initial error."* Thus, the mechanisms for finite predictability within the L63 and L69 model are not exactly the same, although both models contains a saddle point.

Based on the L69 linear model with a saturation assumption, the conceptual model for a chain process possesses a collection of unstable modes that sequentially appear, grow, and saturate. Although the L69 model effectively describes the phenomena of instability, it cannot precisely reveal the true nature of chaos. In contrast to growing solutions, the L69 linear model also produces decaying components (with positive eigenvalues) and oscillatory components (with negative eigenvalues). As documented in [11,12,15,16], the L63 model and its generalized version (e.g., [11]) can produce non-chaotic solutions and coexisting chaotic and non-chaotic solutions. Therefore, a realistic model should possess both chaotic and non-chaotic processes (as well as both unstable and stable solutions) and, thus, distinct predictability. An estimate of a predictability limit using either the (classical) L63 or L69 model, with or without additional assumptions (e.g., saturation), should be interpreted with caution and should not be generalized as an upper limit for predictability.

**Supplementary Materials:** Supplementary Materials can be downloaded at: https://www.mdpi.com/article/10.3390/atmos13050753/s1. Table S1 displays the definitions of CDIC and SDIC. Lists

1–12 are provided for the analysis of the L63 model by Palmer et al., 2014 [71], the fundamental local theorem of ODEs, CDIC, Lipschitz constant, CDIC and the Lipschitz constant, SDIC and chaos, a definition of chaos, volume contraction within the L63 model, a definition of the Lyapunov function, existence of the Lyapunov function and global stability within the L63 model for $r \leq 1$. Figures S1–S7 display: a relationship between $c_2$ in [71] and the Lipschitz constant $L$, an illustration of CDIC and SDIC, numerical experiments of control and parallel runs within the L63 model, time-varying local Lyapunov exponents, dependence of predictability on initial conditions, and qualitative description of local predictability on the Lorenz attractor [74].

**Author Contributions:** B.-W.S. designed and performed research; B.-W.S., R.A.P.S. and X.Z. wrote the paper. All authors have read and agreed to the published version of the manuscript.

**Funding:** This research received no external funding.

**Institutional Review Board Statement:** Not applicable.

**Informed Consent Statement:** Not applicable.

**Data Availability Statement:** Not applicable.

**Acknowledgments:** We thank anonymous reviewers, D. Durran, R. Rotunno, and H.-C. Kuo for valuable comments and discussions.

**Conflicts of Interest:** The authors declare no conflict of interest.

## Appendix A. The Lorenz 1963 Model and Its Simplified Systems

Here, the Lorenz 1963 model [1] is rewritten by introducing two additional parameters ($\sigma_1$ and $\epsilon$) as the coefficients of dissipative terms, as follows:

$$\frac{\mathrm{d}X}{\mathrm{d}\tau} = \sigma Y - \sigma_1 X, \tag{A1}$$

$$\frac{\mathrm{d}Y}{\mathrm{d}\tau} = -XZ + rX - \epsilon Y, \tag{A2}$$

$$\frac{\mathrm{d}Z}{\mathrm{d}\tau} = XY - bZ. \tag{A3}$$

Here, $\tau$ is dimensionless time. The above three, first-order ODEs describe the time evolution of three state variables $X$, $Y$, and $Z$. Two parameters, $\sigma$ and $r$, are called the Prandtl number and the normalized Rayleigh number, respectively. The parameter $b$ is a function of the ratio of the horizontal and vertical scales of a convection cell. The term $-bZ$ introduces dissipation. Compared to the classical Lorenz model, Equations (A1)–(A3) introduce two additional parameters, $\sigma_1$ and $\epsilon$, in order to trace the impact of each of the three dissipative terms. When $\sigma_1 = \sigma$ and $\epsilon = 1$, Equations (A1)–(A3) represents the classical L63 model.

By introducing $\Omega = X^2/2 - \sigma Z$, one can transform Equations (A1)–(A3) into the following 2nd- and 1st-order ODEs that produce the same form as the Pedlosky model [31]:

$$\frac{\mathrm{d}^2 X}{\mathrm{d}\tau^2} + (\sigma_1 + \epsilon)\frac{dX}{d\tau} - (\Omega + \sigma r - \sigma_1 \epsilon)X + \frac{X^3}{2} = 0, \tag{A4}$$

$$\frac{d\Omega}{d\tau} + b\Omega + (\sigma_1 - \frac{b}{2})X^2 = 0. \tag{A5}$$

From left to right, the four terms of Equation (A4) represent acceleration, linear dissipation, linear forcing, and a nonlinear cubic restoring force, respectively.

The above system in Equations (A4) and (A5) has been compared to the Pedlosky model for revealing their mathematical universality, identifying two crucial dissipative terms, and illustrating the physical relevance of related findings to predictability problems.

Equations (A4) and (A5) can be simplified into the following systems:

(a) $\sigma_1 = \sigma$ and $\epsilon = 1$ (i.e., the L63 model):

$$\frac{d^2 X}{d\tau^2} + (\sigma + 1)\frac{dX}{d\tau} - (\Omega + \sigma r - \sigma)X + \frac{X^3}{2} = 0, \tag{A6}$$

$$\frac{d\Omega}{d\tau} + b\Omega + (\sigma - \frac{b}{2})X^2 = 0. \tag{A7}$$

The system has a $r_c = \frac{\sigma(\sigma + b + 3)}{\sigma - b - 1} = 24.74$ for the onset of chaos.

(b) $\sigma_1 = \sigma$ and $\epsilon = 0$ (i.e., the simplest Lorenz-type model for chaos):

$$\frac{d^2 X}{d\tau^2} + (\sigma)\frac{dX}{d\tau} - (\Omega + \sigma r)X + \frac{X^3}{2} = 0, \tag{A8}$$

$$\frac{d\Omega}{d\tau} + b\Omega + (\sigma - \frac{b}{2})X^2 = 0. \tag{A9}$$

The system has a $r_c = \frac{\sigma(\sigma + b)}{\sigma - b} = 17.27$ for the onset of chaos.

(c) $\sigma_1 = b = 0$ (an uncoupled 2D system with $d\Omega/d\tau = 0$):

$$\frac{d^2 X}{d\tau^2} + \epsilon\frac{dX}{d\tau} - (\Omega + \sigma r)X + \frac{X^3}{2} = 0. \tag{A10}$$

The above system is briefly analyzed in the main text, yielding spiral sink solutions.

(d) $\sigma_1 = b = \epsilon = 0$ (i.e., the non-dissipative L63 model with $d\Omega/d\tau = 0$):

$$\frac{d^2 X}{d\tau^2} - (\Omega + \sigma r)X + \frac{X^3}{2} = 0. \tag{A11}$$

As discussed in Shen, 2018, 2020 [29,30] and reviewed in the main text, the above system produces two types of oscillatory solutions and homoclinic orbits.

(e) No nonlinear term in Equation (A11) (i.e., a linear system with $d\Omega/d\tau = 0$):

$$\frac{d^2 X}{d\tau^2} - (\Omega + \sigma r)X = 0. \tag{A12}$$

The above system represents the most fundamental 2nd order ODE with stable and unstable solutions.

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
