# Peer review of "One Saddle Point and Two Types of Sensitivities within the Lorenz 1963 and 1969 Models"

_atmosphere, doi:10.3390/atmos13050753_

Round 1

Reviewer 1 Report (Previous Reviewer 1)

The author have focused on two famous models Lorenz 1963 and 1969. In face they have a review on both models and their mathematical and physical relationship. This is intresting study that can be published in the journal. 

At Page 11, the authors described about some ill-conditioning and they used the condition number. I have some questions! You have a value 10^{-4} in Line 467. What the role of this value? In fact you apply this value to control the accuracy of the results. How do you know that 10^{-4} is a suitable value for your approximation? Why you do not choose for example 10^{-10} or a large value for example 10^{-1}? 

In fact, for small values you will have so many iterations (without improving the accuracy) to find the solution and for large values, you will have 1-2 iterations without having acuurate results. How do you want tosolve this problem? How do you want to find an optimal value to control the accuracy? 

I propose a major revision! 

Author Response

Please see the attached pdf file.

Reviewer 2 Report (Previous Reviewer 4)

None

Author Response

Please see the attached pdf file.

Reviewer 3 Report (New Reviewer)

SUMMARY OF THE PAPER

The Lorenz-63 and -69 models have both been used throughout the literature to make statements on the predictability time limit of the atmosphere. In this paper the authors argue that such statements need to be interpreted with caution. It is shown that the mechanisms leading to finite predictability are different in both models, but that they also have a common feature in the form of a saddle point that produces instability in both systems.

RECOMMENDATION

Unfortunately, I cannot recommend the manuscript in its present form for publication. In my view a major revision would be needed in which the authors address the major comments below.

I have read the manuscript as a mathematician with an interest and background in weather and climate models. So most of my comments below are related to mathematical aspects, which, in my view, should be clarified before I would be able to recommend the manuscript for publication.

MAJOR COMMENTS

1) It is not clear at all to me why equation (1) should have any relation with the Lorenz-63 model. Appendix A provides some details on the derivation, but in my view the rationale is missing in some steps.

1a) On line 631, the authors introduce a new quantity \Omega = X^2/2 - \sigma Z. It is not explained why this quantity is of interest. First of all, \Omega is not an invertible function of X, Y, and Z. So in transforming equations (A1)-(A3) to (A4)-(A5) information is lost. So what can we infer from equations (A4)-(A5) about the original Lorenz-63 model?

Is there perhaps a *physical* reason to consider the quantity \Omega? Why was this specific form chosen? Why not pick \Omega = Y^2 / 2 - \sigma Z, or even something else? The choice of \Omega feels arbitrary, and the authors should explain the rationale behind this choice.

1b) Further assumptions lead to even more simplified equations (A11) and (A12). Again, in what way (both qualitatively and quantitatively) can we learn things from these equations about the Lorenz-63 model?

If it is merely the point that equation (A12) has both stable and unstable solutions and that the origin in the L63 model has both contracting and expanding directions, then why bother doing this derivation? Why not just start with equation (A12) and explain that it shares the feature of having a saddle point with the L63 model? To me it feels that the derivation of equation (A12) has little significance in explaining the point that the authors are trying to make, and it feels to be a very artificial way of trying to explain the relationship between the L63 system and equation (A12).

Perhaps I am missing the point here. But I do think that the authors should explain the rationale more carefully. I consider this to be the weakest part of the manuscript. I think the authors should either provide physical reasons or geometrical reasons (e.g. related to orbit structure) to justify their the derivation of equations (A11) and (A12).

2) Lines 387-390: "However, as discussed below, this limitation does not have a significant impact on our major conclusion since the 9 × 9 matrices represent sub-matrices of the original 21 × 21 matrices at larger scales. Therefore, the eigenvalues are relatively small as compared to the eigenvalues of the corresponding full matrices."

I do not understand this statement. In general, one cannot make any statement on the relationship between the eigenvalues of a matrix and the eigenvalues of a smaller submatrix. If the matrix has a very special structure, then such statements would be possible, but in that case the authors need to explain this.

3) Lines 404-406: "On the other hand, the saturation assumption has been applied to extend a system’s predictability as follows: when the unstable mode at the smallest scale reaches its saturated value, as determined by the energy spectrum, it is removed from the system."

I do not understand this procedure. How are modes removed from the system? Is it done by considering an eigenvector V of a saturated mode and then replacing the matrix A by the matrix P^T A P, where P is the orthogonal projection onto the orthogonal complement of the space spanned by the vector V? Please clarify the procedure of removing modes.

4) The discussion of equation (14) is puzzling to me. The way the equation is written suggests that it is a finite sum. Indeed, equation (4) is a 2N-dimensional system and hence has at most 2N modes. Therefore, I would expect that the sum in equation (14) has at most 2N terms, the last term being 1 / \gamma_{1 \times 1}. Only in infinite-dimensional systems (e.g. dynamical models governed by partial differential equations rather than finitely many ordinary differential equations) I would expect the sum in (14) to have infinitely many terms.

The discussion that starts at line 527 is about infinite series. How does this relate this discussion to the L69 system (which is finite-dimensional and for which T has only finitely many terms)? It seems that the discussion actually is related to the PDE model in equation (3).

The remark "Therefore, whether (or not) Eq. (14) produces a finite predictability is still a challenging question." (lines 540-541) is puzzling to me. Is this about the infinite-dimensional L69 model in equation (3), as a I suspect, or is it really about the 2N-dimensional model of equation (4) (which would not make sense).

MINOR COMMENTS

1) Upon first reading, the paragraph in lines 174-187 was very confusing to me. The first sentence "The linearization is based..." made me wonder which equation was to be linearized. Indeed, equation (4) is already linear. Later I realized that this sentence referred to the *derivation* of the L69 model. So perhaps the authors could emphasize that.

2) Equation (10): in the right hand side the matrix A is typeset in ordinary font, whereas boldface is used in the remainder of the paper. Please be consistent in notation.

3) Does equation (14) have a typo? Should the last term be 1 / \gamma_{7 \times 7}?

Author Response

Please see the attached pdf file.

Round 2

Reviewer 1 Report (Previous Reviewer 1)

I checked the paper again. I do not have more comments on this paper. 

Reviewer 3 Report (New Reviewer)

I would like to thank the authors for their detailed reply to my comments and the changes they have made to the manuscript. In my view, the revised manuscript can be accepted for publication.

This manuscript is a resubmission of an earlier submission. The following is a list of the peer review reports and author responses from that submission.

Round 1

Reviewer 1 Report

In this paper the authors have focused on the Lorenz 1963 and 1969 models. They have described the models and discussed the stability analysis. I have the following comments on this paper: 

1- The introduction can be improved by adding the advantages of your study and disadvantages of previous studies. 

2- The paper is really long with 59 pages. The authors can remove some unnecessary parts. 

3- The paper is not in the format of a paper. It looks like a thesis that the authors added an introduction and abstract. 

4-  There is no special methodology in the paper. 

5- The results of Table 1 are not good! 

6- Thee are some easy and famous relations to find the condition number and also in appendix there are some easy and famous examples. It can make me sure that this is a master thesis, not a professional paper!

I can not recommend the paper for publication.   

Author Response

Please check the PDF (attached).

Reviewer 2 Report

The central issue I have with the paper is that it is just much too long. Exploring these always very well studied systems over almost 40 pages (including appendix), makes its readability suffer. The authors should focus on one or two core messages and findings that they think are the most important and novel findings of their study. The Lorenz 63 in particular is a model that has been studied so often and intensively that for me as a reviewer, it is very hard to judge what really are novel findings and what not (as many parts of the article are also just reproducing known properties). Please cut down the paper to its most important parts and make your core findings really stick out the reader. In the state as it is, it is not a good or interesting read. 

1) In the introduction it says: “As a result, when real-world global models produced encouraging simulations at extended-range (15-30 day) time scales (Shen et al. 2010, 2011; Shen 2019b; Judt 2020), some have interpreted such results as inconsistent with chaos theory.” (Without any reference)

I have a bit of an issue with that remark. Predictability of chaotic systems comes down to its Lyapunov spectrum. This is something that Lorenz also demonstrated in his seminal work. His work demonstrated this _conceptually_ on models that only model the atmosphere in rather crude fashion, they are not detailed models of Earth’s atmosphere. Estimating the predictably of Earth’s atmosphere then is possible by estimating the Lypanuov spectrum of the best possible representation of the Earth’s atmosphere, so some comprehensive Earth system model, as has been done in some studies (or by using methods that can estimate Lyapunov exponents directly from data). That the Lyapunov spectrum (or just the maximum Lyapunov exponent) of such models is different to that of Lorenz’ models is no surprise and definitely not inconsistent with chaos theory. They are just very different models. [I am not implying that the authors write that this is the case, but they state that some people do/did without reference]

2) This leads me to a general question about the paper: you are speaking about predictability of the L63 and L69 multiple times throughout the paper. Why are never investigating or referring to its Lyapanov spectrum?

3) “While an unstable mode may be physically interesting, in the real world, a stable mode likely represents a more realistic solution in response to ini- tial small-scale perturbations. “

Just a comment to that: The authors are of course right that the stable modes are more realistic solutions, however the unstable modes of climate models can be of importance in the real work as well. For an article on that see e.g arXiv:1903.08348  

4) “[…] By comparison, when the Runge-Kutta scheme is applied, numerical solutions within the 2nd column of Table 1 increase with time, indicating an instability […]”

Did you use a Runge Kutta with adaptive or fixed step width? In the case the latter is used, does the result change if an adaptive step width solver is used? 

5) Regarding “SDIC Within the Full or Forced, Non-dissipative L63 Model”

The difference between the two trajectories grows exponentially, I think it would be more instructing to show the logarithm of the difference. However, the chaotic behaviour of the L63 has been shown so often in previous studies and textbooks that I really find it quite unnecessary to show this again in this already quite lengthy study. 

6) “Namely, the predictability limit is determined before the onset of chaos, and the predictability horizon displays a dependence on initial conditions that are close to or away from one of three critical points within the L63 model. “

What exactly do you mean with onset of chaos here? And in which exactly cases, in your view is the predictability dependent on the ICs?

7) “A Conceptual Model for a Chain Process” 

What’s the motivation for setting up this chain process? It’s not clear to me what you are trying to achieve with this. Also it is not clear to me from the description how the process actually looks like. Could you write it in a mathematical description?  

Author Response

Please check the PDF (attached).

Reviewer 3 Report

The paper deals with the comparison of two types of Lorenz equations. Although the content itself is understandable, I do NOT recommend this paper for publication.

The main reason is that I could not really grasp what is the novelty of the paper, what is the objective of the paper and what is the take home message to the readers. During the reading of the manuscript, it was quite frustrating that for pages after pages I had to read decades old, obvious and well-know features of nonlinear dynamical systems. Only a single example: the features of a saddle and its numerical consequences. It is to be stressed, it is only a single example, there are many more.

Author Response

Please check the PDF (attached).

Reviewer 4 Report

This work presents some investigation into the Lorenz (1963) and (1969) predictability models. The main goal of this study is to find some connection between the two models, which supposedly then allows the authors to represent these models in the same mathematical form. With such a unified model and approach, the authors claim to find two types of initial condition sensitivities in L63 and L69’s model, which highlight the different chaotic nature of the two models.

I found the manuscript hard to follow, unorganized, and most of all filled with many unclear/inaccurate descriptions of L63 and L69 models. I will not debunk every such single claim but focus instead on the main idea of this work. Recall first that the L63 model presents a group of dynamical systems that possess a chaotic attractor, which dictates the system predictability based on several conditions 1) the existence of a bounded attractor, 2) the attractor is statistically dense, and 3) the sensitive dependence of the model state on initial condition (i.e., there exists a positive Lyapunov exponent). The limited predictability is understood in the sense that improving an initial condition error by, say, 10 times, would only lengthen the time that the error would occupy the entire attractor and become indistinguishable by a factor of ln(10) ~ 2.5 times. Unlike this deterministic predictability concept in L63’s model, L69’s predictability framework is entirely different in the sense that 1) L69’s model is based on multiscale systems that are homogenous and isotropic, and 2) the error spectral growth is proceeding by the “upscale growth” processes. L69’s predictability framework is therefore applied only to a fully-developed turbulence system that possesses a specific background spectrum. Such a distinct difference in the error growth and saturation between L63’s and L69’s model is so fundamental that Palmer et al. (2014) named the L69’s framework as a “real butterfly effect” to distinguish from L63’s predictability approach/interpretation. Note also the upscale spectral error growth in a multiscale system with limited predictability is much more effective than the linear growth by Lyapunov exponent (see Vallis, Chapter 8 for a more in-depth discussion of such spectral error growth and estimations).

Unfortunately, the authors of this work misinterpret completely this fundamental difference between L63 and L69’s predictability framework when they attempt to unify the two models under the same approach. This misinterpretation of L63 and L69’s work leads to not only inaccurate claims throughout the work but also an unexpected development of their Eq. (1), which is completely different from the original L63’s model (both physically and mathematically), or irrelevant eigenvalue analyses of Eq. (4) that make up a big chunk of this work. I am very surprised to see that the authors repeatedly cite the work by Rotunno and Snyder (2007) as a source of their calculation, yet the fundamental concept of predictability in L69’s model is not even correctly interpreted in this study. Whether or not a multiscale model displays limited predictability depends on the background error spectrum that may or may not allow the upscale growth to be sufficiently fast as seen in Rotunno and Snyder‘s Figures 1 and 2. It is fine that the authors look for the eigenvalues of the A matrix in Eq. (4) to see what critical points are stable as in this work. However, assigning the limited predictability in L69’s to the growth of numerical errors or instability of saddle points as in this study (lines 477-486) is not what L69’s predictability framework is about. For this reason, I don’t see much value of this work, especially for a Special Issue on such Lorenz’s seminal studies.

Palmer et al. 2014: https://doi.org/10.1088/0951-7715/27/9/R123 

Vallis, G. K. 2017:  https://doi.org/10.1017/9781107588417

Author Response

Please check the PDF (attached).
